# ACCIDENTAL EXPLORATION THROUGH VALUE PREDICTORS

## ABSTRACT

Infinite length of trajectories is an almost universal assumption in the theoretical foundations of reinforcement learning. In practice learning occurs on finite trajectories. In this paper we examine a specific result of this disparity, namely a strong bias of the time-bounded Every-visit Monte Carlo value estimator. This manifests as a vastly different learning dynamic for algorithms that use value predictors, including encouraging or discouraging exploration.

We investigate these claims theoretically for a one dimensional random walk, and empirically on a number of simple environments. We use GAE as an algorithm involving a value predictor and evolution strategies as a reference point.

## 1 INTRODUCTION

In practice all reinforcement learning environments produce only trajectories of finite length. In the case of computer simulations this is sometimes hard-coded in the environment, but even when it is not we have to introduce a cutoff to apply most interesting algorithms. Due to this cutoff we are forced to estimate the value function using the time-bounded Every-visit Monte Carlo estimator. As we will show this estimator is significantly biased.

The time bound bias we will investigate stems directly from omitting the suffix of an infinite trajectory in the calculation of the value of a state. Because of that the influence of this bias is stronger in states that tend to appear close to the end of a trajectory. We use this asymmetry in experiments to investigate the bias by examining its influence on exploration for agents that use value predictors.

### 1.1 INFLUENCE ON EXPLORATION

To find the optimal solution an agent needs to explore, that is, visit a significant portion of the state space, even if locally the reward does not encourage this behaviour. Exploration plays a fundamental role in reinforcement learning, and ensuring that algorithms explore sufficiently remains a pressing research problem. Exploration can either be undirected or directed (Thrun, 1992). The former refers to methods like $\epsilon$-greedy or softmax strategies (Tijsma et al., 2016) and its effectiveness is very limited. The latter attempts to implement some form of intrinsic motivation, usually based on success rate when attempting to predict future states, with methods ranging from visit counts (Tang et al., 2017) to prediction networks (Anonymous, 2019).

To show the connection of the time bound bias with exploration consider the influence of the bias on a given state $s$. If, on average, the missing suffix of the total discounted reward in $s$ is positive (resp. negative), then the bias lowers (resp. increases) the learned value of $s$. The later the state $s$ appears in trajectories, the stronger the bias. Thus, if in a region of the state space all the missing suffixes have similar magnitude and are positive (resp. negative), then the bias discourages (resp. encourages) the agent to visit states that appear at later points in trajectories.

It would seem natural that negative rewards should therefore encourage exploration – states that are, on average, visited later, usually lie in previously unvisited parts of the latent space. Reaching them quicker might allow the agent to go even further. Perhaps the simplest environment in which this is the case is a constant trajectory length variant of the MountainCar environment (Moore, 1990) from the OpenAI gym (Brockman et al., 2016). In this environment all rewards received before reaching the target – the single absorbing state – are equal to $-1$, and zero afterwards. Exploration

is a crucial component of solving the problem. We will also show that positive rewards strongly discourage exploration and in some cases the agent simply learns to stay in proximity to the starting state.

However, negative rewards are not a solution to the exploration problem – we will construct an environment where rewards are all equal to $-1$, but the preference on states introduced by the bias actually discourages exploration. We also propose some ways of mitigating the bias in general.

## 1.2 RELATED WORK

This paper investigates issues related to the problem of learning good policies for general Markov Decision Processes with access to only finite experience. A general algorithm for finding a near-optimal solution of a MDP in polynomial time has been given in Kearns & Singh (2002). This algorithm is model-based, so it can generate constant length simulations of the environment starting at any state. In our model-free setting, in contrast, the length of future experience available for a given state depends on the time it is visited. As described in the previous section this introduces some undesirable behaviour when using the Every-visit Monte Carlo estimator and our goal is to analyse these issues.

## 2 THE TIME-BOUND BIAS

We will use the following notation. When an agent using policy $\pi$ experiences a trajectory

$$(s_1, a_1, r_1, s_2, \ldots) \sim \pi$$

the total discounted reward is

$$G_t := \sum_{k=t}^{\infty} \gamma^{k-t} r_k.$$

The value function is defined as

$$V_\pi(s) := \mathbb{E}_\pi [G_t \mid s_t = s]. \tag{1}$$

In the Every-visit Monte Carlo estimator the expected value in equation 1 is replaced by an average over all occurrences of the state $s$ within $N$ sampled trajectories. Denoting the number of times the state $s$ is visited in the $j$-th trajectory by $M_j$ we have:

$$\hat{V}_{\pi,N}(s) := \frac{\sum_{j=1}^{N} \sum_{t:\, s_{j,t}=s} G_{j,t}}{\sum_{j=1}^{N} M_j}. \tag{2}$$

The average is ill defined for states that occur an infinite number of times in any of the $N$ trajectories, but the common assumption is that the only such state is a single absorbing state. We assume that this state has constant reward $r_a$ and its discounted value is $\frac{r_a}{1-\gamma}$. In Sutton et al. (1998) the authors have shown that the Every-visit Monte Carlo estimator has a bias which tends to $0$ as $N$ tends to infinity. This is in contrast to the unbiased First-visit Monte Carlo estimator.

In this paper we make the more realistic assumption that every trajectory's length is equal to a constant value $T$. In such a case we use the time-bounded version of the total discounted reward:

$$G_t^T := \sum_{k=t}^{T} \gamma^{k-t} r_k.$$

The time-bounded Every-visit Monte Carlo estimator is therefore

$$\hat{V}_{\pi,N}^T(s) := \frac{\sum_{j=1}^{N} \sum_{t:\, s_{j,t}=s} G_{j,t}^T}{\sum_{j=1}^{N} M_j^T}, \tag{3}$$

where $M_j^T$ counts the occurrences of state $s$ in the truncated trajectory.

Finally, we define the time bound bias as the difference between the expected value of the time-bounded Every-visit Monte Carlo estimator and the value function in equation 1

$$\text{T-bias}_{\pi,N}(s) := \mathbb{E}_\pi \left[ \hat{V}_{\pi,N}^T(s) \right] - V_\pi(s). \tag{4}$$

Unlike the case of the unbounded Every-visit Monte Carlo estimator this bias does not disappear when $N$ tends to infinity. To show the reason let us define $p_\pi(s_t = s)$ as the probability that for a random trajectory sampled using policy $\pi$ the $t$-th state is equal to $s$. Then, by dividing the numerator and denominator of equation equation 3 by $N$ and changing the order of summation we get

$$\hat{V}_{\pi,N}^T(s) = \frac{\sum_{t=1}^T \frac{1}{N} \sum_{j\,:\,s_{j,t}=s} G_{j,t}^T}{\frac{1}{N} \sum_{j=1}^N M_j^T}.$$

By plugging this into equation equation 4 and taking the limit with respect to $N$, we define

$$\text{T-bias}_\pi(s) := \lim_{N\to\infty} \text{T-bias}_{\pi,N}(s) = \frac{\sum_{t=1}^T p(s_t = s)\, \mathbb{E}_\pi\left[G_t^T \mid s_t = s\right]}{\sum_{t=1}^T p(s_t = s)} - V_\pi(s). \tag{5}$$

The last part of equation equation 5 would be equal to 0 if we replaced all $\mathbb{E}_\pi\left[G_t^T \mid s_t = s\right]$ with $\mathbb{E}_\pi\left[G_t \mid s_t = s\right]$. Therefore this bias stems directly from the missing suffixes of the trajectories.

We can reach similar conclusions for the First-visit Monte Carlo estimator – both $M_j^T$ and $p_\pi(s_t = s)$ would have to be redefined to be the number of first visits and probability that $s_t$ is the first visit respectively. This estimator will also from suffer from a variant of the time bound bias, as the problem with missing suffixes of trajectories persists. In this paper we focus only on the Every-visit variant, as it is more popular.

Finally, let us define

$$V_\pi^T(s) := \frac{\sum_{t=1}^T p(s_t = s)\, \mathbb{E}_\pi\left[G_t^T \mid s_t = s\right]}{\sum_{t=1}^T p(s_t = s)}, \tag{6}$$

so that we simply have $\text{T-bias}_\pi(s) = V_\pi^T(s) - V_\pi(s)$.

If we try to learn the value function, we encounter three obstacles – the time bound bias, the bias described in (Sutton et al., 1998), and limitations in how well the model learns. In our paper we assume the model succeeds at approximating the value estimator. We limit the approximation error by using simple environments with low dimensional state spaces and training the value predictor until it converges. We ensure the second bias is low by using a large $N$. Thus, the effects we describe are mainly caused by $\text{T-bias}_\pi$.

## 3 RANDOM WALK

In this section we investigate a simple theoretical model, namely a time-bounded one-dimensional random walk with constant reward $r$. Let $s \in \{\bot\} \cup ([-B, B] \cap \mathbb{Z})$ be the state, and $t \in \{1, \ldots, T\}$ – the time. The agent's policy $\pi$ is constant and defined as $(p_{-1}, p_0, p_{+1})$, where $p_a$ is the probability of moving from state $s$ to state $s + a$. Attempting to visit the integers $B + 1$ or $-B - 1$ causes instant transmission to the absorbing state $\bot$. Then the agent receives an infinite stream of rewards equal to $r$.

In the case of an infinite random walk the value function is constant and equal to $V_\pi(s) = \frac{r}{1-\gamma}$. A $T$-step random walk illustrates how an agent experiences truncated trajectories of the infinite case. We will analyse the function $V_\pi^T$, which here is equal to the $\text{T-bias}_\pi$ up to a constant factor. The walk starts at $s = 0$, thus $|s| \le T$. We also assume that $T < B$. As a consequence, the time-bounded agent will never reach the boundaries or the absorbing state, so we do not have to treat them in any special way in the calculations below.

We define the probability distribution $p_\pi(s, t)$ to be the probability of the state being $s$ and the time being $t$. We may assume that the marginal distribution $p_\pi(t)$ is uniform and equal to $\frac{1}{T}$ – this is convenient, and consistent with the definition of $V_\pi^T$ if we say that $p_\pi(s_t = s) = p_\pi(s \mid t)$

$$p_\pi(s, t) = p_\pi(s \mid t) \cdot p_\pi(t) = \frac{p_\pi(s \mid t)}{T}.$$

It is worth noting that in our experiments this distribution can be interpreted as the distribution of data from the point of view of the value predictor. In this case the marginal distribution on time is also uniform, but only because we use Every-visit Monte Carlo as our value estimator.

We can calculate $p_\pi(s, t)$ using the recurrent formula for $p_\pi(s \mid t)$:

$$p_\pi(s \mid t = 1) = \begin{cases} 1 & \text{if } s = 0, \\ 0 & \text{otherwise,} \end{cases}$$

$$p_\pi(s \mid t = k + 1) = p_{+1} \cdot p_\pi(s - 1 \mid t = k) + p_0 \cdot p_\pi(s \mid t = k) + p_{-1} \cdot p_\pi(s + 1 \mid t = k),$$

and now we have:

$$G_t^T = r \frac{\gamma^{T-t+1} - 1}{\gamma - 1},$$

$$V_\pi^T(s) = \frac{\sum_{t=1}^T p_\pi(s, t) \cdot G_t^T}{\sum_{t=1}^T p_\pi(s, t)}.$$

For examples of the value function with $r = 1$ see Figure 1.

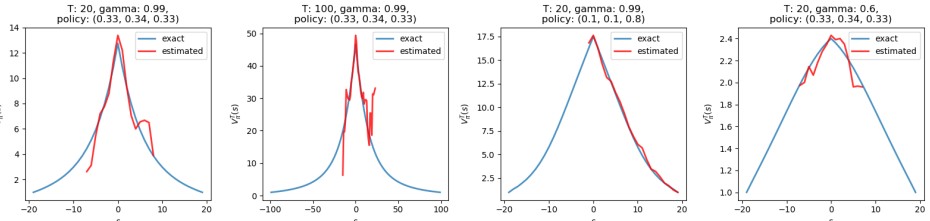

Figure 1: Value functions for a finite discrete random walk. Note that the vertical axis varies between the 4 plots.

As it turns out, the value function is an even function if only $p_{-1}, p_{+1} > 0$. To see this note that every partial trajectory that ends at time $t$ in state $s$ can be bijectively coupled with a mirror partial trajectory, where every step right is replaced with step left and step left replaced with step right. Thus if $p_{-1}, p_{+1} > 0$, then $p(s, t) = (\frac{p_{+1}}{p_{-1}})^s p(-s, t)$, and we get

$$V_\pi^T(-s) = \frac{\sum_{t=1}^T p_\pi(-s, t) \cdot G_t^T}{\sum_{t=1}^T p_\pi(-s, t)} = \frac{\sum_{t=1}^T (\frac{p_{+1}}{p_{-1}})^{-s} p_\pi(s, t) \cdot G_t^T}{\sum_{t=1}^T (\frac{p_{+1}}{p_{-1}})^{-s} p_\pi(s, t)} = \frac{\sum_{t=1}^T p_\pi(s, t) \cdot G_t^T}{\sum_{t=1}^T p_\pi(s, t)} = V_\pi^T(s).$$

The value function is linear with respect to the constant reward $r$. For $r = 1$ we will observe that the value function has a sharp peak at $0$. The optimal policy that maximises the average value of a visited state is to always stay at $0$. Thus, generalised policy iteration will be strongly biased towards this particular policy. Learning using an advantage function instead of the reward will have a similar effect, as in the case of constant reward the preference stems only from the value function. If we set $r = -1$ the effects will be symmetric and in turn encourage moving away from the beginning of the walk.

In Appendix A we perform a detailed analysis of the even simpler case of the Wiener process with no discount factor. This might provide the reader with a better intuition regarding the bias, as it is a good approximation of the discrete random walk.

## 3.1 BIAS PREVALENCE

The time bound bias is a result of choosing the time-bound Every-visit Monte Carlo estimator, so it will be present in any setting when such a choice is made. The discrete random walk is a very simple example, but it illustrates the main properties of the bias that we are concerned with, that is the influence on exploration. We argue the bias will also have similar properties in other environments for several reasons.

First of all, the time bound bias is continuous with respect to the reward and policy. As long as rewards are of a similar order of magnitude and are either usually positive or usually negative, the value estimator should end up with a sharp peak (either positive or negative) close to the starting state. Moreover, the time bound bias is local in the sense that it does not require the whole state

space to have appropriate properties. As long as an agent ends up in a previously unexplored region of the state space that exhibits the required properties, the value estimator of the new states should be biased in a similar way, encouraging or discouraging exploration in this region.

In the following sections we will carry out experiments, and empirically test the impact of the T-bias. We will also introduce certain modifications to the environments, and discuss the resulting changes in agent's performance in the context of the bias.

## 4 EXPERIMENTAL SETUP

The complete codebase used for this paper can be found in the Github repository [anonymised for review] Here we just describe the basics of our experimental setup.

### 4.1 ALGORITHMS

We use three algorithms throughout the paper, with various modifications that are described where applicable. Those three are:

**Generalized Advantage Estimation** also referred to as GAE throughout the paper, as described in Schulman et al. (2016). In short this algorithm uses two neural networks, one to estimate the value of a state (referred to as the *value predictor*), and another to actually take the actions (referred to as the *agent*). We train the value predictor to minimise the difference between predicted value and the estimator defined by equation equation 3, which is one of the approaches discussed in the original paper.

**Evolution Strategies** as described in Rechenberg (1973). This is a vary basic algorithm, which nonetheless presents excellent exploration in the problems considered. We use it for reference as an algorithm that achieves very good performance in the simple environments presented here.

**Policy Gradient** as described in Williams (1992). This algorithm does not estimate values of states and the effects described in this paper do not occur. Despite being basic, it can significantly outperform algorithms with value predictors if the T-bias degrades their performance.

### 4.2 NETWORKS

All the networks used in the experiments have the same architecture, including the value predictor. They consist of two hidden affine layers of size $64$, each followed by an application of a leaky ReLU with leak $0.1$. The final layer is again affine, with the output size dependent on the purpose of the net. The agents are all stochastic, so their networks (i.e. all but the value predictor) end with an application of softmax. The weights in the networks are initialised with values close to $0$ using a normal distribution.

### 4.3 LEARNING PROCESS

In the case of the evolution algorithm and policy gradient the learning process is straightforward, we just run a single episode of the environment under consideration and immediately update the agent using the gathered experience. With GAE we update the agent when the first episode is completed after taking $2048$ steps. This lowers the bias due to a finite number of trajectories when using the Every-visit Monte Carlo value estimator. By default we use a discount factor of $0.98$. In all cases we use the Adam optimiser (Kingma & Ba, 2015).

GAE needs one more parameter called $\lambda$, controlling the impact of advantages with different time gaps between states. Our theoretical discussion essentially assumes $\lambda = 0$, that is computing the advantages only using values in consecutive steps. However, the original GAE paper (Schulman et al., 2016) suggests using much bigger values and most implementations follow their advice. In our experiments setting $\lambda = 0.95$ did not add significant effects, so to keep our results practical we use this value.

## 4.4 NORMALISATION

To provide greater stability we normalise the cumulative discounted rewards passed to the policy gradient and evolution agents, and also the advantages passed to the GAE agent. We use a running normalise with a discount factor of $\frac{1}{2}$. We do not perform any normalisation of target values for the value predictor.

# 5 ENVIRONMENTS

We consider two environments to show the effects described in section 3. Later we explore various modifications of those environments to pinpoint the exact nature of the effects.

All the graphs presented in this and further sections contain translucent dots, representing the total episode rewards in specific runs of the algorithms, and lines representing the average reward. We ran 16 copies of every algorithm. When interpreting those one should take note that while the lines move, dots usually appear in the same areas. This is due to the challenge in the environments being mostly exploration, so whenever an algorithm finds any solution it often quickly learns how to achieve it consistently.

## 5.1 MOUNTAINCAR

First we consider the MountainCar environment from the OpenAI gym. A single episode ends after at most 200 steps. This is equivalent to saying that we set $T = 200$ and after reaching the target, the agent enters the absorbing state and receives rewards equal to zero afterwards. Because of the sparse reward in this environment any algorithm not including a nontrivial exploration component gets stuck close to the starting position. In practice no trajectory will randomly reach the goal, so there simply is no data to learn from. Yet, when ran with the GAE algorithm, the environment gets solved quite rapidly. The results are shown in Figure 2a.

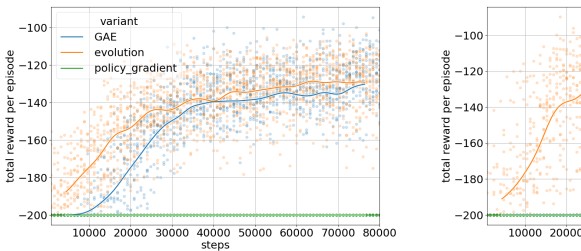

(a) The standard MountainCar environment     (b) The modified DragCar environment

Figure 2: Performance of three algorithms on both the Car problems.

This is precisely the result we should expect. Since all the steps have reward $-1$ and all random trajectories have length 200, the states that are visited on average later in a trajectory have a higher value estimation. It remains to note that a random trajectory starts close to the centre of the valley, but will gather some momentum with time, thus making the states further from the centre appear more valuable. With a value predictor the agent will be therefore incentivised to seek out states further from the centre, which eventually leads it to the goal.

## 5.2 AXIS WALK

To show that the time bound bias can also have obviously negative consequences we consider a very simple environment, in which the agent walks on the axis of integers, starting at 0. In every step it is allowed to move either right or left and the reward at a step ending up in state $s$ (which is an integer) is $1 - \frac{1}{8(1+s^2)}$. Every episode ends in the same number of steps, either 20 or 200, depending on the variant of the environment used. It is easy to see that, since states further from the centre have higher rewards, the optimal strategy is to pick a direction and always walk that way.

This environment is a very close approximation of the random walk described in Section 3, since all rewards are quite close to 1. Therefore we should expect the GAE agent to avoid states far from 0, which is the exact opposite of the optimal strategy. Our expectations are confirmed in Figure 3. The effect is weaker in the variant with more steps and we will discuss this phenomenon in Section 6.

Note that here even the policy gradient performs well in comparison to the GAE. This task is extremely easy, but the T-bias makes it impossible to solve, in reasonable time, using algorithms relying on value predictors.

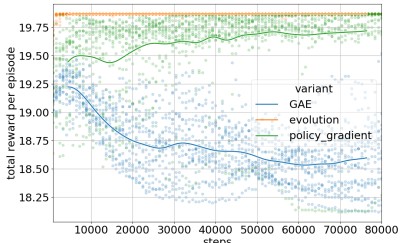
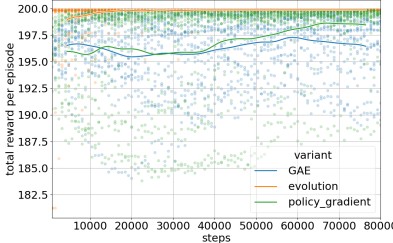

(a) Axis walk with 20 steps per episode      (b) Axis walk with 200 steps per episode

Figure 3: Performance of three algorithms on two variants of axis walk.

## 5.3 DRAGCAR

The above examples might give the impression, that just setting rewards in every step to a constant negative value encourages exploration. To dispel this notion we present a slight modification of MountainCar in which standard value predictors discourage exploration. This requires two modifications. First, one needs to introduce drag to the environment. The drag coefficient is not substantial and decreases the maximal achievable reward insignificantly. It's especially worth pointing out that just adding drag makes the algorithms perform about as good as without it.

Second, one needs to move the starting position. Instead of starting in the middle of the valley, the car randomly starts either halfway up the left or halfway up the right hill. Due to the drag, if the agent does nothing the car will slowly converge to the middle of the valley. So will in fact a random agent. It will obviously still create some swinging, but it's amplitude will be significantly smaller than the difference between the starting points. In this case the value predictor cannot extrapolate good rewards uphill. When this happens the agent often learns to stay at the bottom of the valley, never leaving the spot. This is confirmed by the results visible on Figure 2b, which presents a sharp drop in the performance of GAE compared to Figure 2a, as it never reaches the goal. Also note that the evolution strategy performs better on this problem than on the standard MountainCar. This is due to the fact that the car starts with greater potential energy.

## 5.4 MOUNTAINCAR+R

To see the scope of the effects described above it is useful to look at another set of slight modifications of the MountainCar environment. Observe that adding a constant value $R$ to every reward does not change the goal of the (infinite) environment – the optimal policy is still to reach the absorbing state in as few steps as possible. Note that the value $V_\pi$ of every state increases by $\frac{R}{1-\gamma}$. This also includes the absorbing state, as now the agent receives infinite stream of rewards $R$ after reaching the top of the hill.

The OpenAI implementation of the MountainCar environment treats it as an episodic task, which ends immediately after reaching the target state. To fit it into our paradigm, we extend all trajectories to be of infinite length, and treat the goal state as an absorbing state. The agent cannot leave it, and all steps after reaching it give reward zero. We set $T = 200$. Next, we produce two new environments by setting $R$ to either 1 or 2. We will call them MountainCar+1 and MountainCar+2 respectively. Even if the time limit T was a part of the task, and not just a restriction on the trajectory length, these

modifications would not change the goal or the optimal strategy of an agent. They do, however, change the behaviour of GAE agents, precisely because of the time bound bias.

After adding 1, all the rewards in a trajectory that does not reach the goal state are 0. In such a case the value estimator $V_\pi^T$ is nearly 0, so the impact of the bias on exploration is negligible. Adding 2 on the other hand makes all the rewards 1, so early states are more valuable, which discourages exploration. The results can be seen in Figure 4. Note that the performance of the evolution strategy remains unchanged.

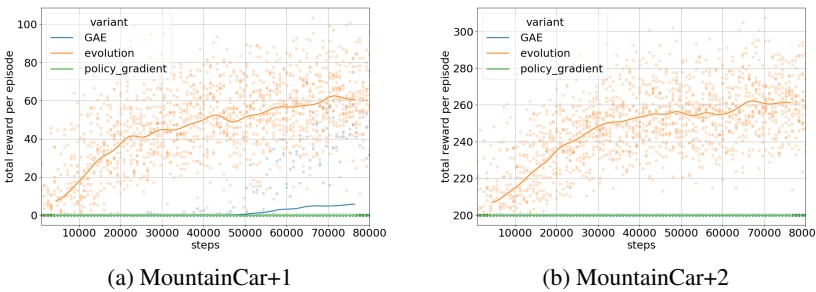

(a) MountainCar+1  (b) MountainCar+2

Figure 4: Performance of all the algorithms on the modified MountainCar environment, as described in Section 5.4.

In the case of MountainCar+1 we can still see some exploration. We argue that this is mainly due to the "consistent noise", by which we mean that a randomly initialised value predictor will assign higher value to some areas in state space, so the GAE agent will be attracted to those areas. We discuss this phenomenon briefly in Appendix C. Nevertheless, we see that its impact on exploration is significantly weaker.

## 6 MITIGATING EFFECTS OF THE BIAS

In this section we propose three attempts at removing effects caused by the time bound bias. None of the solutions are completely satisfactory, but each one approaches the problem from a different angle and partially explains the relationship between the bias and exploration.

### 6.1 ADJUSTING $\gamma$

We can limit the bias by lowering the discount factor $\gamma$. When we do so, rewards included in the missing suffixes of the trajectories become more discounted, so removing them introduces weaker bias. Note that lengthening the episode has a similar effect, as mentioned in Section 5.2. The results of adjusting the discount factor can be seen in Figure 5.

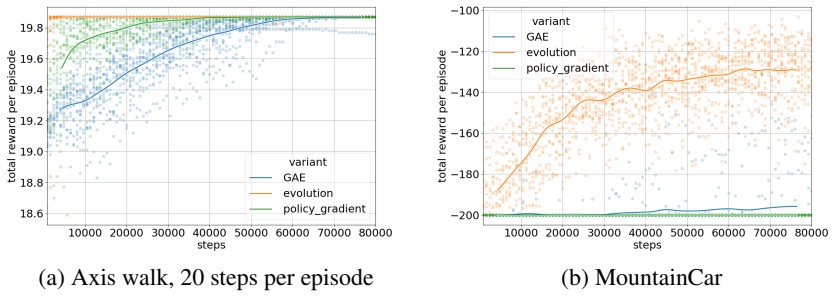

(a) Axis walk, 20 steps per episode  (b) MountainCar

Figure 5: Performance when we lower the discount factor in GAE from $0.98$ to $0.8$.

Of course this does not completely eliminate the biases, they are still influential near the ends of the trajectories. What is perhaps a more important problem with this solution is the obvious trade-off. Lowering the discount factor actually makes the agents less sensitive to rewards further away in the future. In the case of many environments this is not an acceptable sacrifice.

## 6.2 BETTER APPROXIMATION OF REWARDS AFTER EPISODE END

The second class of solutions we propose focuses on finding a better guess for what the rewards after time $T$ should be. We discuss some approaches, but once again none are quite satisfactory. A very simple idea would be to assume the value of the last state is the discounted average reward summed to infinity.[1] This approach introduces a new bias, but in our environments it works reasonably well in practice, see Figure 6.

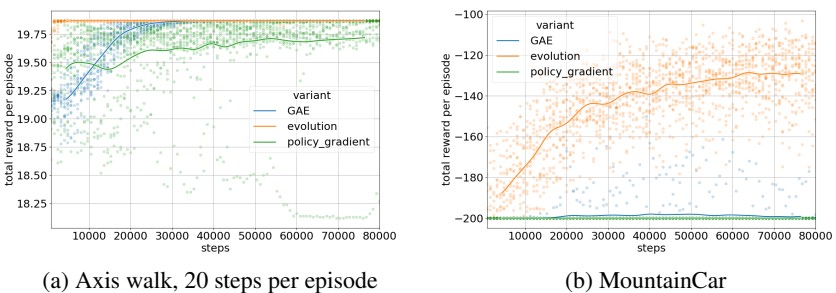

(a) Axis walk, 20 steps per episode    (b) MountainCar

Figure 6: Performance when the value of the last state is assumed to be the average reward summed to infinity.

Another approach would be to use the value predictor to simply predict the value of the last state and assume this prediction is correct. The assumption introduces some reflective dependency on the learning process, which is of course worrying because of the added complexity. This proves to be a problem also in practice, see Figure 7. In particular note that the biases are all still present, but at least in the case of Axis walk the modification eventually helps. The prediction of the value for an absorbing state is still a problem, since if we cannot distinguish it from other states we cannot set it to $0$, and it keeps being determined by the state of the value predictor.

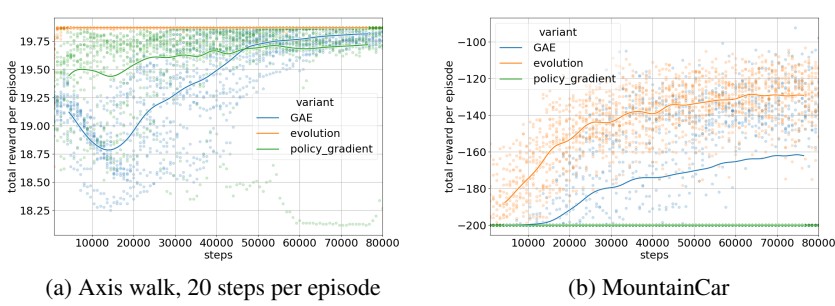

(a) Axis walk, 20 steps per episode    (b) MountainCar

Figure 7: Performance where the value of the last state is assumed to be correctly predicted by GAE.

## 6.3 ENFORCING THE MARKOV PROPERTY

The final approach is to present a different perspective on the effects' origin. Let us observe that cutting off trajectories early is equivalent to stating that we modify the environment's definition, so that after time $T$ the agent enters an absorbing state, and all consecutive rewards are equal to zero.

---

[1]We want to thank [name redacted] for bringing this idea to our attention.

These two descriptions are indistinguishable from the agent's perspective, thus all effects observed during training must be explainable in the latter case. Note that this modification does not change the optimal policies of the environments we present in this work.

In the modified environment the time-bound bias vanishes completely, but the time-dependence of the last transition violates the definition of a Markov Decision Process. Therefore, we can attempt to remove the undesirable effects by reintroducing the Markov property. We did it by making one final modification: we append time to states. The results can be seen in Figure 8. The remaining exploration in the case of MountainCar is mostly caused by the consistent noise effect. In particular, compare this to Figure 4a.

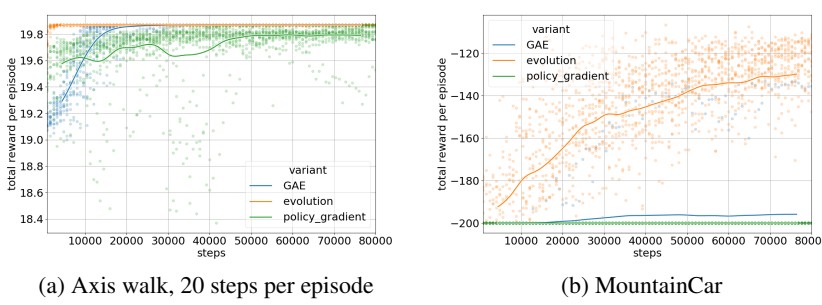

(a) Axis walk, 20 steps per episode      (b) MountainCar

Figure 8: Performance when time is part of the state.

This solution lets us gain an important insight into the problem, but is unsatisfactory, because the environment represents a completely different, although related, process.

Another way of removing the problematic effects would be, instead of using a "hard" cutoff for trajectory length, to allow the trajectory to randomly enter the absorbing state at any time with constant probability $p$. This approach might be an interesting area of research, as it might provide a better way of representing infinite Markov decision processes in finite implementations. However, it violates our assumption about trajectories of constant length, so it is beyond the scope of this paper.

## 7 CONCLUSIONS

In many environments using value predictors might result in unexpected learning patterns. We demonstrated that both discouragement and encouragement of exploration, as well as constraining movement to a small segment of the state space, occur in simple environments. When using GAE or similar algorithms one should therefore exercise caution, keep in mind the possibility of bias, and where applicable attempt some form of mitigation. We proposed a couple partial solutions together with advise when to use or avoid them.

Value predictors are of course very useful and work quite well in many settings, so despite the flaws we demonstrated they should enjoy continued popularity. We hope that further research will discover either better ways of mitigating the biases described in this paper or some algorithms that avoid the problems altogether while keeping the strengths of value predictors.

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

## A    WIENER PROCESS

We will use the following properties of a Wiener process:

1. A Wiener process $W_t$ is a family of probability distributions on $\mathbb{R}$.
2. $W_0 = 0$ almost surely.
3. $W_t \sim \mathcal{N}(0, t)$.

Since it is a normal distribution, $W_t$ has density function $f_t(x) = \frac{1}{\sqrt{2\pi t}} exp(-\frac{x^2}{2t})$. Constraining ourselves to the time-bounded case translates directly into the assumption that $t \in [0, T]$. If we further assume that the time is distributed uniformly, we can define the density function on $\mathbb{R} \times (0, T]$:

$$f(x, t) = \frac{1}{T} \frac{1}{\sqrt{2\pi t}} exp\left(-\frac{x^2}{2t}\right).$$

We now can define the average time at the state $x$, denoted by $T(x)$. This value is of interest to us, because in the discrete case with constant reward $r$ in every step, visiting state $x$ at time $t$ results in (undiscounted) cumulative reward equal to $r(T - t)$. In that case the average value of state $x$ is equal to $r(T - \mathcal{T}(x))$. The average time at the state $x$ is

$$\mathcal{T}(x) = \frac{\int_0^T t f(x, t) dt}{\int_0^T f(x, t) dt} = -\frac{x^2}{3} + \frac{T}{3} \frac{1}{1 - \sqrt{\pi} \frac{|x|}{\sqrt{2T}} exp(x^2/2T) \operatorname{erfc}(|x|/\sqrt{2T})},$$

where erfc is the *complementary error function*:

$$\operatorname{erfc}(x) = \frac{2}{\sqrt{\pi}} \int_x^\infty e^{-y^2} dy. \tag{7}$$

See appendix B for detailed calculations.

For clarity, we may write

$$\mathcal{T}(-x\sqrt{2T}) = \mathcal{T}(x\sqrt{2T}) = \frac{T}{3}\left(-2x^2 + \frac{1}{1 - \sqrt{\pi}x \cdot exp(x^2) \operatorname{erfc}(x)}\right)$$

for $x \geq 0$. This cannot be expressed in terms of elementary functions, but the part $\sqrt{\pi}x \cdot exp(x^2) \operatorname{erfc}(x)$ can be approximated with small relative error, see Oldham (1968).

Even though erfc is not an elementary function we know that $\operatorname{erfc}(0) = 1$ and $(\operatorname{erfc}(x))' = \frac{2}{\pi} exp(-x^2)$. Thus we may calculate the exact value of $\mathcal{T}$ and its directional derivatives at zero. For example $\mathcal{T}(0) = \frac{T}{3}$ and $\mathcal{T}'(0) = \sqrt{\frac{T\pi}{18}}$. This gives some intuition about the steepness of the function in the proximity of the beginning of the walk. The value at zero grows linearly with $T$, while the steepness – sublinearly. The expected time for several trajectory lengths can be seen in Figure 9.

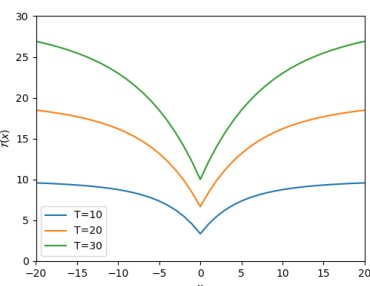

Figure 9: Expected time at a state, for several trajectory lengths.

Setting $W_t \sim \mathcal{N}(0, t\sigma^2)$ for $\sigma > 0$ entails $f_{\sigma^2}(x, t) = \frac{1}{\sigma} f(\frac{x}{\sigma}, t)$ and, as a consequence, $\mathcal{T}_\sigma(x) = \mathcal{T}(x/\sigma)$. This should remind us of the discrete $N$-step random walk, where a single step has mean zero and variance $\sigma^2$. For large $t$ the probability distribution of states in such a walk converge in distribution to $\mathcal{N}(0, t\sigma^2)$ due to the central limit theorem.

## B   THE AVERAGE TIME AT A STATE IN THE WIENER PROCESS

In this appendix we compute the average time at a state in the Wiener process. First some definitions relating to the error function:

$$\text{erf}(x) = \frac{2}{\sqrt{\pi}} \int_0^x e^{-y^2} dy,$$

$$\text{erfc}(x) = 1 - \text{erf}(x) = \frac{2}{\sqrt{\pi}} \int_x^\infty e^{-y^2} dy,$$

$$\text{erf}(+\infty) = 1.$$

We will also use the following integrals:

$$\int \frac{e^{-x^2}}{x^2} dx = -\frac{e^{-x^2}}{x} - \sqrt{\pi}\,\text{erf}(x), \tag{8}$$

$$\int \frac{e^{-x^2}}{x^4} dx = \frac{1}{3}\left(-\frac{e^{-x^2}}{x^3} + 2\frac{e^{-x^2}}{x} + 2\sqrt{\pi}\,\text{erf}(x)\right). \tag{9}$$

We want to calculate the expected time at state $x$

$$\mathcal{T}(x) = \frac{\int_0^T t f(x,t) dt}{\int_0^T f(x,t) dt},$$

where

$$f(x,t) = \frac{1}{T} \frac{1}{\sqrt{2\pi t}} exp(-\frac{x^2}{2t}).$$

First we compute the denominator:

$$\int_0^T f(x,t) dt = \int_0^T \frac{1}{T} \frac{1}{\sqrt{2\pi t}} exp(-\frac{x^2}{2t}) dt = \ldots$$

$$\left[w^2 = \frac{x^2}{2t},\ dt = -\frac{x^2}{w^3} dw,\ \sqrt{t} = \frac{|x|}{w\sqrt{2}}\right]$$

$$\ldots = \int_{+\infty}^{\frac{|x|}{\sqrt{2T}}} -\frac{1}{T} \frac{w}{|x|\sqrt{\pi}} \frac{x^2}{w^3} e^{-w^2} dw$$

$$= \int_{+\infty}^{|x|/\sqrt{2T}} -\frac{1}{T} \frac{|x|}{w^2\sqrt{\pi}} e^{-w^2} dw$$

$$= \frac{1}{T} \frac{|x|}{\sqrt{\pi}} \int_{\frac{|x|}{\sqrt{2T}}}^{+\infty} \frac{e^{-w^2}}{w^2} dw \tag{10}$$

$$\overset{(8)}{=} \frac{1}{T} \frac{|x|}{\sqrt{\pi}} \left(-\frac{e^{-w^2}}{w} - \sqrt{\pi}\,\text{erf}(w)\right)\Big|_{\frac{|x|}{\sqrt{2T}}}^{+\infty}$$

$$= \frac{1}{T} \frac{|x|}{\sqrt{\pi}} (0 + e^{-x^2/(2T)} \frac{\sqrt{2T}}{|x|} - \sqrt{\pi}(1 - \text{erf}(\frac{|x|}{\sqrt{2T}})))$$

$$= \frac{1}{T} (\sqrt{\frac{2T}{\pi}} e^{-x^2/(2T)} - |x|\,\text{erfc}(\frac{|x|}{\sqrt{2T}})).$$

Now the numerator:

$$
\begin{aligned}
\int_0^T t f(x,t)dt &= \int_0^T t \frac{1}{T} \frac{1}{\sqrt{2\pi t}} exp(-\frac{x^2}{2t})dt = \dots \\
&\left[ w^2 = \frac{x^2}{2t}, \ dt = -\frac{x^2}{w^3}dw, \ \sqrt{t} = \frac{|x|}{w\sqrt{2}} \right] \\
\dots &= \frac{1}{T} \frac{|x|^3}{2\sqrt{\pi}} \int_{\frac{|x|}{\sqrt{2T}}}^{+\infty} \frac{e^{-w^2}}{w^4}dw \\
&\overset{(9)}{=} \frac{1}{T} \frac{|x|^3}{2\sqrt{\pi}} \frac{1}{3} (-\frac{e^{-x^2}}{x^3} + 2\frac{e^{-x^2}}{x} + 2\sqrt{\pi}\,\mathrm{erf}(x)) \Big|_{\frac{|x|}{\sqrt{2T}}}^{+\infty} \\
&= \frac{1}{T} \frac{|x|^3}{2\sqrt{\pi}} \frac{1}{3} (0 + e^{x^2/(2T)}((\frac{\sqrt{2T}}{|x|})^3 - 2\frac{\sqrt{2T}}{|x|}) + 2\sqrt{\pi}(1 - \mathrm{erf}(\frac{|x|}{\sqrt{2T}}))) \\
&= \frac{1}{T}(\frac{T-x^2}{3}\sqrt{\frac{2T}{\pi}}e^{-x^2/(2T)} + \frac{x^2}{3}|x|\,\mathrm{erfc}(\frac{|x|}{\sqrt{2T}}))
\end{aligned}
\tag{11}
$$

Thus, combining the results:

$$
\begin{aligned}
\mathcal{T}(x) &= \frac{\int_0^T t f(x,t)dt}{\int_0^T f(x,t)dt} \\
&\overset{(10)(11)}{=} \frac{\frac{1}{T}(\frac{T-x^2}{3}\sqrt{\frac{2T}{\pi}}e^{-x^2/(2T)} + \frac{x^2}{3}|x|\,\mathrm{erfc}(\frac{|x|}{\sqrt{2T}}))}{\frac{1}{T}(\sqrt{\frac{2T}{\pi}}e^{-x^2/(2T)} - |x|\,\mathrm{erfc}(\frac{|x|}{\sqrt{2T}}))} \\
&= -\frac{x^2}{3} + \frac{T}{3}\frac{\sqrt{\frac{2T}{\pi}}e^{-x^2/(2T)}}{\sqrt{\frac{2T}{\pi}}e^{-x^2/(2T)} - |x|\,\mathrm{erfc}(\frac{|x|}{\sqrt{2T}})} \\
&= -\frac{x^2}{3} + \frac{T}{3}\frac{1}{1 - \sqrt{\pi}\frac{|x|}{\sqrt{2T}}e^{x^2/(2T)}\,\mathrm{erfc}(\frac{|x|}{\sqrt{2T}})}.
\end{aligned}
$$

## C  BIAS OF THE VALUE PREDICTOR

### C.1  CONSISTENT NOISE

Here we discuss the phenomenon shown in Figure 4a. In case of the MountainCar+1 environment the agent receives constant reward equal to zero until it reaches the absorbing state. It turns out that the GAE agent sometimes manages to solve the environment, despite having no exploration boost caused by the T-bias – the Monte-Carlo value estimator is equal to zero.

We argue that the main source of exploration is the value predictor, which is being initialised randomly and therefore containing some bias. By "consistent noise" we mean that a randomly initialised value predictor will assign higher value to some areas in state space, so the GAE agent will be attracted to those areas. Since the predictor is updated relatively slowly, this attraction will be somewhat consistent between episodes. Also, the advantages are normalised before passing them to the agent. If the value network is trained to predict zero everywhere, then even small deviations from the target will be magnified after the normalisation.

We further validate this explanation by showing that the standard MountainCar environment can be solved by a variant of the GAE algorithm, which does not update the value predictor (see Figure 10). Note that the results are comparable to the MountainCar+1.

### C.2  OPTIMISTIC INITIALISATION

It is valuable to compare the experiments with the MountainCar environment to the experiments in Sutton et al. (1998). There the source of exploration is the biased initialisation of (tabular) value

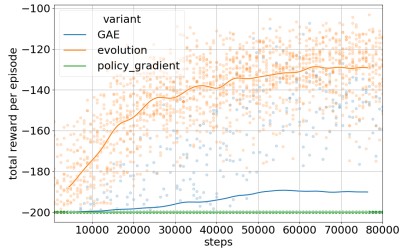

Figure 10: Performance of the GAE algorithm on the MountainCar environment when the value predictor is never updated.

predictor. All states have initial value equal to zero, and then the algorithm learns that the visited states actually have negative value, thus tries to explore the unseen parts of the state space.

This kind of bias is weaker if the value predictor is a neural network, as it can generalise to new regions of the state space. Also, it does not help to solve the DragCar environment. On the other hand, the T-bias has much stronger influence, because it can boost the exploration even on the already seen part of the state space, and not only on the borders of the known region.

