# OpenReview forum: "Accidental exploration through value predictors"
_ICLR.cc/2019/Conference_

### Official Review · AnonReviewer2 · 2018-10-14
**not formal enough**

**Rating:** 3
**Confidence:** 4

**Review:**


UPDATE:

I have read the authors’ response and the other reviews.  While the authors have made some improvements, my core criticism remains – the paper does not produce concrete theoretical or empirical results that definitively address the problems described.  In addition, there are many confusing statements throughout the paper.  For instance, the discussion of positive and negative rewards in the introduction does not conform with the rest of the literature on exploration in RL.

The authors also seemed to have missed the point of the Kearns & Singh reference.  The authors are right that the older paper is a model-based approach, but the idea is that they too were solving infinite-horizon MDPs with finite trajectories and not introducing a bias.


Summary:

The paper attempts to link the known mismatch between infinite horizon MDP values and finite trajectory sums to the problem of exploration.  Trajectories in environments requiring exploration (mountain car and a number-line walk) are shown and the effects of changing trajectory lengths and initial values are discussed.  Potential solutions to the problem are proposed though the authors did not deem any of the solutions satisfactory.


Review:

The paper brings up a number of important issues in empirical reinforcement learning and exploration, but fails to tackle them in a manner that convincingly isolates the problem nor proposes a solution that seems to adequately address the issue.  Specifically, several issues seem to be studied at once here (including finite-horizon MDPs, function approximation, and exploration), relevant work from the exploration and RL community is not cited, the early experiments do not reach a formal theoretical claim, and the proposed solutions do not appear to adequately address the problem).  These issues are detailed below.

First, the paper is considering many different issues and biases at once, including those introduced by initialization of the value function, exploration policies, function approximation, and finite/infinite length trajectories.  While the authors claim in several places that they show one bias is more important than another, no definitive experiment or theorem is given showing that finite-length trajectories are the cause of bad behavior.  While it is well known that infinite-horizon MDPs do not exactly match value functions for finite horizon MDPs, so many other factors are included in the current work (for instance the use of neural networks) that it remains unclear that the finite/infinite mismatch is an issue.

The paper also fails to cite much of the relevant work on these topics.  For instance, the use of infinite-horizon MDPs to study finite learning trajectories is often done under the guise of epsilon-optimal guarantees, with  epsilon derived from the discount factor (see “Near-Optimal Reinforcement Learning in Polynomial Time”).  In addition, the effects shown in mountain car when changing the values or the initialization function, mirror experiments with Q-learning that have shown that there is no one initialization scheme that guarantees optimal exploration (see Strehl’s thesis “Probably Approximate Correct Exploration in Reinforcement Learning” ).  Overall, the paper seems to confuse the problems of value initialization and trajectory length and does not show that they are particularly related.

In addition, the early sections covering theoretical models such as Wiener Processes and Random Walks lay out many equations but do not come to a specific formally proven point.  No theorem or proof is given that compactly describes which exact problem the authors have uncovered.  Therefore, when the solutions are presented, it remains unclear if any of them actually solve the problem.

Finally, several of the references are only ArXiv pre-prints.  Papers submitted to ICLR or other conferences and journals should only cite papers that have been peer-reviewed unless absolutely necessary (e.g. companion papers).

---

> ### Author Response · Authors · 2018-11-23
> **Response**
>
> We thank the reviewer very much for their comments, they were very helpful in improving our
> work. We revised the paper to make it more formal and focused on the issue we
> want to present. In a newly added chapter 2 The time-bound bias we provide a formal definition of the bias and a clear statement of the problem.
>
> “First, the paper is considering many different issues and biases at once, including those introduced by initialization of the value function, exploration policies, function approximation, and finite/infinite length trajectories. While the authors claim in several places that they show one bias is more important than another, no definitive experiment or theorem is given showing that finite-length trajectories are the cause of bad behavior. While it is well known that infinite-horizon MDPs do not exactly match value functions for finite horizon MDPs, so many other factors are included in the current work (for instance the use of neural networks) that it remains unclear that the finite/infinite mismatch is an issue.”
>
> Our only goal is to analyse the effects of the time-bound bias. The last paragraph of chapter 2 explains which other biases we need to take into consideration. There, we describe the steps we’ve taken to ensure that they are negligible in our setting - i.e. after every update of the agent the value network is trained long enough to reflect the value estimation produced by the Every-visit Monte Carlo estimator.
> We could replace GAE with tabular methods, but we think it is quite fun to show that GAE fails to solve one-dimensional gridworld with truncated trajectories, in contrast to the policy gradient.
>
> “The paper also fails to cite much of the relevant work on these topics. For instance, the use of infinite-horizon MDPs to study finite learning trajectories is often done under the guise of epsilon-optimal guarantees, with epsilon derived from the discount factor (see “Near-Optimal Reinforcement Learning in Polynomial Time”). In addition, the effects shown in mountain car when changing the values or the initialization function, mirror experiments with Q-learning that have shown that there is no one initialization scheme that guarantees optimal exploration (see Strehl’s thesis “Probably Approximate Correct Exploration in Reinforcement Learning” ). Overall, the paper seems to confuse the problems of value initialization and trajectory length and does not show that they are particularly related.”
>
> We have introduced a short discussion of the related work, including the article “Near-Optimal Reinforcement Learning in Polynomial Time”. We are particularly grateful for pointing out this paper, because it gave us an opportunity to underline what the focus of our work is.
> The reviewer’s remark about initialisation of the value network is, if we understand correctly, a result of confusion caused by lack of clarity in the submitted version of our paper. We apologize for this. The only time we want to discuss value network bias is the case of the MountainCar+1 environment, where the effects caused by the time-bound bias should be nonexistent, but the agent still (but rarely) explores. We try to explain this phenomena in terms of the value network bias, but this is unrelated to the main concern of this paper - we moved the discussion to the Appendix. This is also why we don’t reference the paper concerning initialisation schemes.
>
> “In addition, the early sections covering theoretical models such as Wiener Processes and Random Walks lay out many equations but do not come to a specific formally proven point. No theorem or proof is given that compactly describes which exact problem the authors have uncovered. Therefore, when the solutions are presented, it remains unclear if any of them actually solve the problem.”
>
> In a newly added subsection 3.1 Bias prevalence we connect the results of discrete random walk analysis with other environments. We argue that since the time bound bias is continuous with respect to the reward and policy, then as long as rewards are of a similar order of magnitude and are either usually positive or usually negative, the value estimator should end up with a sharp peak close to the starting state, thus resulting in exploration boost or discouragement. The rest of the paper tries to check these claims empirically, and explain the mechanism.
>
> “Finally, several of the references are only ArXiv pre-prints. Papers submitted to ICLR or other conferences and journals should only cite papers that have been peer-reviewed unless absolutely necessary (e.g. companion papers).”
>
> We also thank the reviewer for pointing out the ArXiv citations -- two of the papers cited that way were actually accepted to previous editions of ICLR, but we failed to cite them appropriately. In the current version the only pre-print we cite is the OpenAI Gym whitepaper.

---

### Official Review · AnonReviewer1 · 2018-10-17
**Papers suffers from confounding of concepts, requires overhaul; in this state it is clearly not publishable.**

**Rating:** 5
**Confidence:** 4

**Review:**

The paper addresses the problem of truncating trajectories in Reinforcement Learning. The scope is right for ICLR.

The presentation is pretty good as far as the English goes but suffers from serious problems on the level of formulating concepts. Overall, I believe that the issue addressed by the paper is important, but I am not sure whether the approach taken by the authors addresses it.

I have the following complaints.
1. The paper suffers from a fundamental confusion about what problem it is trying to address. There are two straightforward ways to formulate the problem. First, we can formulate the task as solving an MDP with infinite trajectories and ask the question of what we can learn training from finite ones. The learned policies would then be evaluated on the original MDP (in practice using trajectories that are, say, an order of magnitude longer or, for a simple MDP, analytically). Second, we can consider the family of episodic MDPs parametrised by trajectory length T and ask the question of what we can learn by training on some values of T and evaluating on others. These problems are similar, but not the same and should be carefully distinguished. Right now, the introduction reads like the authors were trying to use the first approach but Section 4 reads like they are doing the second: "If during training we just used times up to T, but deployed the agent in an environment with times greater than T". Either way, the concept of bias, which appears throughout the paper, isn't formally defined anywhere. I believe that a paper that claims to address bias should have an equation that defines it as a difference between two clearly defined quantities. It should also be clearly and formally distinguished which quantities are deterministic and which are random variables. The introduction seems to (implicitly?) define bias as a random variable, section 3.1 seems to talk about "bias introduced by initializing the network". As the paper stands now, the working definition of bias used by the authors seems to be that some quantity is vaguely wrong. I do expect a higher standard of clarity in a scientific paper,

2. The paper confounds the problem of learning the value function, specifying the initial estimates of the value function and exploration. The analysis of exploration is entirely informal and suffers from the lack of clear problem formulation as per (1). Of course, one can influence exploration by initialising the value function in various ways, and this may respond differently to different truncations (different values of T), but I don't see how it is related to the "bias" problem that the paper is trying to address. In any case, I wish the authors either provided a formal handle on exploration or shift the focus of the paper and remove it,

3. I don't see what hypothesis the experiments are trying to test. Clearly, if I train my agent on a different MDP and test it on a different one, I get a mismatch. The lack of clear definitions as per (1) comes back with a vengeance.

4. Section 1.1 seems to exist purely to create a spurious impression of formality, which bears little relevance for what the paper is actually about. RL, as traditionally formulated, uses discrete time-steps so the Brownian motion model developed in this section doesn't seem very applicable - it is true that it is a limiting case of a family of discrete chains, but I don't see how this produces any insights about RL - chains are easy to simulate, so why not test on a chain directly? In any case, the result shown in Section 1.1 is entirely standard, can be found in any textbook on stochastic processes and was likely introduced purely to cover for the lack on any substantial theory that comes form the authors.

To summarize: if the authors address these points, there is a possibility that the ideas presented in this draft may somehow lead to a paper at some later point. However, I feel that the required changes would be pretty massive and don't see how the authors could make it during the ICLR revision phase - the problems aren't details or technicalities but touch the very substance of what the paper is trying to do. Basically, the whole paper would need to be re-written.

another minor point:  sloppy capitalization in section 2.1

===================

For reasons outlined in my comment below, I updated the score to 5 (for the new heavily updated version).

---

> ### Author Response · Authors · 2018-11-23
> **Response**
>
> We would like to thank the reviewer very much for their comments, they were extremely helpful in improving our work.
>
> To address the complaints:
>
> 1. We edited the paper to make the concepts more clear. In a newly added chapter 2. The time-bound bias we provide, among others, a formal definition of the bias.
> We always assume that we are trying to solve an infinite MDP, while learning on trajectories of length T. We’ve removed all the confusing parts and further clarified in chapter 2.
>
> 2. Our only goal is to analyse the effects of the time-bound bias. The last paragraph of chapter 2 explains which other biases we need to take into consideration. We argue that they are negligible in our setting. The "bias introduced by initializing the network" becomes relevant only if we significantly lessen the time-bound bias - a short discussion about the value network biases was moved to the Appendix C.
>
> 3. We trained and tested all the agents on the same environments. After each modification of the environment the agent and the value network were trained again from scratch. The experiments show the influence of the time-bound bias on exploration - this is why we modify the environments slightly, and compare the changes in the agent’s policy (after retraining) with those we would expect to happen in the infinite-time case. For example, adding a constant value to all rewards in the MountainCar environment does not change the objective nor the optimal policy, but the GAE agent behaves completely differently. The figures show only the history of training, but in our opinion this is sufficient to illustrate the effects we investigate.
>
> 4. “Section 1.1 seems to exist purely to create a spurious impression of formality (...) and was likely introduced purely to cover for the lack on any substantial theory that comes from the authors.”
> This remark is particularly hard for us to address. We will try our best to explain the reasons behind introducing the Wiener process analysis, and let the reviewer judge our true intentions.
> The continuous case is a good approximation of the discrete random walk, and we are able to explain why the bias generates a sharp peak by calculating the one-sided derivative at the beginning of the walk. We simply think that is an even more basic example, and the final equations provide additional insight.
> On the other hand, we fully agree that this chapter is redundant, and disrupts the paper’s flow, so we decided to move it to the Appendix.
>
> As the reviewer suggested, to improve clarity of the paper changes in the structure needed to be quite massive, but we, hopefully, managed to address all the mentioned problems, without the need to modify the experiments or the conclusions. We are fully aware that this fact alone is sufficient to disqualify our submission, nevertheless we would be extremely grateful for any further comments.

---

> > ### Comment · AnonReviewer1 · 2018-11-25
> > **Update after reading revised version.**
> >
> > I wanted to thank the authors for going through the process of revising the paper. The changes they have made are substantial and do reflect the most important issues I mentioned in the review. The clarity of the presentation is greatly improved and the concept of time-induced bias has now been given a clear definition. The experiments are better described and the section on the chain MDP fits much better with the rest of the paper.
> > My concerns are:
> > 1. The paper focuses on the setting where the episode length is very short (for long episodes and realistic values of gamma, the bias will still exist but will be negligible). It would be good if the authors provided a domain that motivated a setting where generating long trajectories is expensive nu generating many episodes is cheap. This is not the case in standard RL domains, where the assumption is that the cost of simulation is roughly the same for each state action pair. However, domains like this probably do exist (think of a situation where the cost of simulation increases in trajectory length).
> > 2. For realistic values of gamma, gamma^n = eps implies n=682 for typical values of gamma=0.99 and  eps = 0.001. I.e. the effect of a reward diminishes to almost zero after circa 682 steps. This suggests a baseline – ignore the last n steps of the trajectory while doing MC estimation. Of course this will only work if the T > n and every state is visited for some t < n in an infinite number of episodes. This point is related to the remark of AnonReviewer2
> > 3. Another baseline that the paper misses is using the learned value of the state in place of the rollout when approaching the end of trajectory. This assumes that we have something to bootstrap the values off, i.e. that every state appears in the initial part of the episode for infinitely many episodes.
> > 4. The experimental validation is still pretty limited, although much better explained. A new domain along the lines of point (1) would be nice.
> > Despite the above, I do appreciate the updates and I raise my score to 5. Because of the above-mentioned concerns, I am still inclined to reject, although the rejection is now marginal. I fully understand if the other reviewers choose to accept. If the paper gets rejected, I do encourage the authors to address the points above and submit it once more to another venue.

---

> > > ### Author Response · Authors · 2018-11-28
> > > **Further response**
> > >
> > > We thank the reviewer for these comments. We are happy to hear that now the paper is more readable.
> > >
> > > 1. Defining an environment in which longer trajectories are more computationally expensive is a very interesting problem. We agree that such environments probably exist -- for example any environment which procedurally generates a world and has to keep it in memory might be a good candidate.
> > >
> > > On the other hand, one could argue that such environments will not be MDPs -- computing the transition from state s should depend only on s and the chosen action, thus the simulation cost can vary for different state-action pairs, but cannot increase in time. Also, if we assume that the process is non-stationary, and modify the agent accordingly, then the effects due to the time-bound bias will probably disappear -- see the experiments in section 6.3, where we include time in the state space.
> > >
> > > It is worth mentioning that using constant-length trajectories that are relatively short is quite popular. For example, consider the OpenAI Gym implementation of the MountainCar environment (T=200), which is for many young researchers one of the first forays into reinforcement learning, and Schulman's et al. experiments with GAE ("Each episode was terminated after 2000 timesteps if the
> > > robot had not reached a terminal state beforehand"). Therefore, we think that our paper is valuable as a straightforward argument against this practice: we introduce the time-bound bias, analyse it on simple environments (where its influence can be isolated), and warn about its ill effects.
> > >
> > > 2. We agree with the reviewer, but with one caveat -- even if the bias has small magnitude, its shape remains the same. Thus, we have to be careful not to amplify the bias later on, e.g. by normalising the advantage, as the effects on exploration might manifest again (in particular, this will happen if a random agent receives constant reward).
> > >
> > > Since rewards after more than n steps have low impact on the state's value, we could simply use n-step rollouts for value estimation, even for the early states. This approach should completely eliminate the effects on exploration, as they stem from using shorter rollouts for later states. Certainly, this
> > > is true for environments with constant reward, as the estimated value will also be constant. We can even choose a relatively small n, and even though the time-bound bias might still be substantial, now it should only be responsible for making the agent prefer instant rewards.
> > >
> > > Perhaps the main disadvantage of this approach is low sample efficiency -- we do not use the last n rewards to update the estimated value of the last n states. Intuitively, it should be possible, but we point out that simple solutions do not work well.
> > >
> > > 3. Possibly we do not understand correctly, but we think we are doing something similar in section 6.2 (figure 7) -- we update the value predictor using sums of the rollouts, and the, properly discounted, current estimation of the last state's value.
> > >
> > > Maybe the reviewer meant n-step bootstrapping for all but the last n states? In such case, we are quite confident this would eliminate the effects on exploration, because they should appear only if we used smaller values of n for later states. This again would have the disadvantage of low sample efficiency for large n.
> > >
> > > Once again we would like to thank the reviewer for their helpful and encouraging comments.

---

> > > > ### Comment · AnonReviewer1 · 2018-11-30
> > > > **Reply to authors.**
> > > >
> > > > Thanks for the reply.
> > > > A couple of remarks:
> > > > 1. The definition of the MDP formalism doesn't explicitly say what the *computational cost* of a transition is, only that the next state is sampled conditioned on the previous one. You can think of settings where the computational cost cost depends on the state. Now, if the transition dynamics is such that the states with more computationally expensive transitions are generated later in the trajectory, long trajectories will be more expensive to generate then a set of shorter trajectories with the same total number of transitions.
> > > > Having a domain like this would be a much more compelling than variants of Mountain Car you currently have, where you have to go from episode length 200 to 20 (a whole order of magnitude) to see the truncation effect. Also, I wouldn't say the horizon truncation problem is an issue with standard RL domains - there, the episode length is set to a number which is large enough to make the bias negligible.
> > > > 2. Practical large-scale implementations of RL mostly use n=batch size anyway.
> > > > 3. Thanks for the clarification.

---

### Official Review · AnonReviewer3 · 2018-11-05

**Rating:** 4
**Confidence:** 4

**Review:**

Paper summary: This paper focuses on the case where the finiteness of trajectories will make the underlying process to lose the Markov property and investigates the claims theoretically for a one-dimensional random walk and Wiener process, and empirically on a number of simple environments.

Comments: The language and the structure of the paper are not on a very good scientific level. The paper should be proofread as it contains a lot of grammatical mistakes.

Given the assumption that every state's reward is fixed, the theoretical analysis is trivial.

The comparison of policy gradient methods is too old. The authors should look for more advanced methods to compare.

The experimental environment is very simple in reinforcement learning tasks, and the authors should look for more complex environments for comparison. The experiment results are hard to interpret.



Q1: In the theoretical analysis, why should the rewards for each state be fixed?

Q2:Why use r_t – (V(s_t)-\gammaV(s_{t+1})) as the advantage function?

Q3: What does the “variant” mean in all figures?

Typos: with lower absolute value -> with lower absolute values

---

> ### Author Response · Authors · 2018-11-23
> **Response**
>
> We thank the reviewer very much for their comments. We edited the paper to make the
> concepts clearer.
>
>
> “The comparison of policy gradient methods is too old.“
> We use the policy gradient mainly to show that, despite being basic, it can still outperform
> algorithms with value predictors if the time-bound bias degrades their performance.
> We also use the ES as a reference algorithm, that performs well and is insensitive to the time-bound bias.
>
> “The experimental environment is very simple in reinforcement learning tasks, and the authors should look for more complex environments for comparison.”
> Using simple environments is the only way to ensure that the observed effects are solely caused by the time-bound bias. At the end of section 3. Random walk we have added a paragraph explaining what other biases we need to consider, and why they’re negligible in this setting.
>
> “The experiment results are hard to interpret.”
> We tried our best to clarify in the current version. Hopefully this makes the results more clear-cut.
>
> Q1:
> This condition was introduced just for simplicity, as varying rewards would lengthen the analysis without helping to convey the core idea. In a newly added subsection 1.1 Influence on exploration we discuss two factors that cause the exploration effects. Firstly, if the missing suffix (after truncating the trajectory to length T) of the total discounted reward in state ‘s’ is positive (resp. negative), then the bias lowers (resp. increases) the learned value of ‘s’. Secondly, the later the state ‘s’ appears in trajectories, the stronger the bias. Combination of these two factors causes the sharp peak in estimated value function.
> Later, in a newly added subsection 3.1 Bias prevalence, we also point out that the effects are local in time, in the sense that if an agent ends up in a previously unexplored region of state space that exhibits the required properties, the value estimator of the new states will be similarly biased.
> Practical effects of non-constant reward are shown in section 5.2 Axis walk.
>
> Q2:
> We tried clarifying this in the revised version. At the end of the section 3 Random walk we say that the optimal policy for the (initially) estimated value is to always stay at 0, therefore the generalized policy iteration will be strongly biased towards this particular policy. Additionally, learning using an advantage function will have a similar effect, as in the case of constant reward the preference stems only from the value function.
> Later we show empirically that the advantage estimator used by GAE also causes the exploration effects.
>
>
> Q3:
> Variant of learning algorithm used in this case -- either the ES, GAE or policy gradient. The colours in the figures correspond to these algorithms.
>
> Once again we thank the reviewer for the comments.

---

### Meta-Review · Area_Chair1 · 2018-12-13
**Please resubmit the paper**

**Confidence:** 4
**Recommendation:** Reject

**Metareview:**

The paper studies the mismatch between value estimation in RL from finite vs. infinite trajectories. This is an interesting problem, but the reviewers raised concerns regarding (1) the consistency and coherence of the story (2) the significance of theoretical analysis and (3) significance of the results. I appreciate that the authors made significant changes to the paper to address the comments. However, given the extent of changes, I think another review cycle is needed to check the details of the paper again.